# Stoichiometry of Rtt109 complexes with Vps75 and histones H3-H4

Noushin Akhavantabib[1], Daniel D Krzizike[2], Victoria Neumann[1], Sheena D'Arcy[1]

**Histone acetylation is one of many posttranslational modifications that affect nucleosome accessibility. Vps75 is a histone chaperone that stimulates Rtt109 acetyltransferase activity toward histones H3-H4 in yeast. In this study, we use sedimentation velocity and light scattering to characterize various Vps75–Rtt109 complexes, both with and without H3-H4. These complexes were previously ill-defined because of protein multivalency and oligomerization. We determine both relative and absolute stoichiometry and define the most pertinent and homogeneous complexes. We show that the Vps75 dimer contains two unequal binding sites for Rtt109, with the weaker binding site being dispensable for H3-H4 acetylation. We further show that the Vps75–Rtt109–(H3-H4) complex is in equilibrium between a 2:1:1 species and a 4:2:2 species. Using a dimerization mutant of H3, we show that this equilibrium is mediated by the four-helix bundle between the two copies of H3. We optimize the purity, yield, and homogeneity of Vps75–Rtt109 complexes and determine optimal conditions for solubility when H3-H4 is added. Our comprehensive biochemical and biophysical approach ultimately defines the large-scale preparation of Vps75–Rtt109–(H3-H4) complexes with precise stoichiometry. This is an essential prerequisite for ongoing high-resolution structural and functional analysis of this important multi-subunit complex.**

## Introduction

The nucleosome is the repeating unit of chromatin and consists of 146 bp of DNA wrapped around a histone octamer. The histone octamer is composed of two copies of histone H2A-H2B dimer and one copy of histone (H3-H4)$_2$ tetramer. Within the octamer, the histones interface through four-helix bundles between the two copies of H3 (H3 and H3'), as well as H4 and H2B (Luger et al, 1997; McGinty & Tan, 2015). Several lysine residues in the histones are acetylated by enzymes that transfer an acetyl group from an acetyl-CoA cofactor. This acetylation modulates the chromatin structure to regulate processes such as DNA replication, repair, and transcription (Shahbazian & Grunstein, 2007; Bannister & Kouzarides, 2011).

In yeast, Rtt109 is an acetyltransferase that modifies various H3-H4 lysine residues, including H3 K9, K27, and K56 (Schneider et al, 2006; Fillingham et al, 2008; Radovani et al, 2013). H3-K56 acetylation is important in fungal pathogenicity, making Rtt109 an attractive antifungal therapeutic target (Wurtele et al, 2010; Dahlin et al, 2014). The position of acetylation is important as it determines the specific downstream effect. H3-K9Ac and H3-K27Ac, for example, influence transcription, whereas H3-K56Ac is additionally associated with DNA replication and repair (Roh et al, 2005; Recht et al, 2006; Chen et al, 2008; Li et al, 2008; Creyghton et al, 2010). The regulation of Rtt109 activity and selectivity is purportedly modulated by two histone chaperones, Vps75 and Asf1 (Driscoll et al, 2007; Tsubota et al, 2008; Kuo et al, 2015). Here, we study the complex between Vps75, Rtt109, and H3-H4.

Vps75 belongs to the nucleosome assembly protein family of histone chaperones (Selth & Svejstrup, 2007). It is composed of two chains that pack together through a head-to-tail coiled coil, with each end of the coiled coil also containing a globular domain (Tang et al, 2008). The Vps75 dimer can also dynamically self-associate to form a homo-tetramer (a dimer of dimers) depending on the protein and salt concentration (Bowman et al, 2014). Vps75 binds H2A-H2B and H3-H4 with nanomolar affinity, and this interaction interferes with Vps75 tetramerization but not with the four-helix bundle interface of (H3-H4)$_2$ (Park et al, 2008; Hammond et al, 2016). To date, the complex between Vps75 and H3-H4 has been refractory to high-resolution structural studies.

Vps75 also binds Rtt109 tightly, and several crystal structures of this complex are available (Albaugh et al, 2010; Su et al, 2011; Tang et al, 2011). These crystal structures show either a 2:1 or a 2:2 complex and identify an Rtt109 loop important for interaction (Su et al, 2011; Tang et al, 2011). Vps75 enhances Rtt109 acetyltransferase ($k_{cat}$ increased ~100-fold) and favors H3-K23Ac and H3-K27Ac (Berndsen et al, 2008; Fillingham et al, 2008; Keck & Pemberton, 2011). This enhancement is 20-fold greater than the enhancement with Asf1, which favors H3-K56Ac (Tsubota et al, 2008). A structural model with both Vps75 and Asf1 suggests these preferences are driven by fuzzy

[1]Department of Chemistry and Biochemistry, The University of Texas at Dallas, Richardson, TX, USA   [2]Department of Biochemistry and Molecular Biology, Colorado State University, Fort Collins, CO, USA

Correspondence: sheena.darcy@utdallas.edu
Daniel D Krzizike's present address is Department of Cancer Biology, Fox Chase Cancer Center, Philadelphia, PA, USA

electrostatic interactions between the negative C-terminal tail of Vps75 and the positive N-terminal tail of H3 (Danilenko et al, 2019).

Our goal is to develop a more complete understanding of Rtt109 regulation by characterizing the complex between Vps75 and Rtt109 both with and without H3-H4. Given the various crystal structures of Vps75 with Rtt109, and the ability of both Vps75 and H3-H4 to oligomerize, we focus on determining relative and absolute stoichiometry. We use various strategies to optimize the purity, yield, homogeneity, and solubility of relevant complexes. Our protocols will facilitate future high-resolution structural approaches. Ultimately, we reveal that Vps75, Rtt109, and H3-H4 form a 4:2:2 complex that is composed of two 2:1:1 sub-complexes. These sub-complexes come together through the four-helix bundle of the (H3-H4)$_2$ tetramer.

## Results

### Vps75 homodimer contains two unequal Rtt109-binding sites

Similar to many proteins in the Nap family, Vps75 adopts dimeric and tetrameric assemblies depending on the ionic strength of the buffer (McBryant & Peersen, 2004; Bowman et al, 2014; Sarkar et al, 2020). Vps75 also interacts with Rtt109 with reported stoichiometries of 2:1 and 2:2 (Su et al, 2011; Tang et al, 2011). To determine the influence of ionic strength on this stoichiometry, we have performed sedimentation velocity analysis with mixtures of individually purified proteins (Table S1). In this setup, we detected Vps75 tetramers and dimers at 5.8 and 4.0 S, respectively (Fig 1A and B, white). We analyzed these Vps75 assemblies with various equivalents of Rtt109. At 150 mM NaCl, addition of 0.5 equivalents did not increase sedimentation, whereas the addition of one equivalent increased sedimentation to 6.7 S (Fig 1A, grey). An increase was not observed at 0.5 equivalents as Rtt109 splits a Vps75 tetramer into two dimers, as shown by others (Hammond et al, 2016). These data suggest that the 6.7 S species is a 2:2 complex at 150 mM NaCl. By contrast, at 300 mM NaCl, addition of 0.5 or 1 equivalent both

increased sedimentation to 5.9 S (Fig 1B, grey). Addition of one equivalent also resulted in the appearance of a tail tending toward 3.2 S, indicative of some unbound Rtt109 (Fig S1A). This shows that the 5.9 S species is a 2:1 complex at 300 mM NaCl. Thus, the Vps75 dimer can bind two copies of Rtt109 at 150 mM NaCl and only one copy at 300 mM NaCl.

To ascertain the absolute stoichiometry of the complexes, we performed size exclusion chromatography coupled to multi-angle light scattering (SEC-MALS) at 150 mM NaCl (Fig 1C). The Vps75 tetramer eluted at 11.9 ml, and Rtt109 eluted at 15.3 ml. Mixtures of Vps75 with 0.5, 1, or 2 equivalents of Rtt109 eluted between these values, reinforcing that Rtt109 binding splits a Vps75 tetramer into two dimers. At 0.5 equivalents, the measured molecular weight (MW) was 117.5 ± 0.07 kD, indicating a 2:1 complex (theoretical MW of 115.7 kD). When Rtt109 is increased to 1.0 and 2.0 equivalents, the MW did increase but remained below a fully saturated 2:2 complex. Unbound Rtt109 was also visible in the chromatogram. These data indicate that while two copies of Rtt109 can bind a Vps75 dimer, the second copy binds with less affinity than the first and is lost during chromatographic separation. Two unequal binding sites were previously suggested by sedimentation equilibrium experiments (Park et al, 2008). A second copy of Rtt109 will not bind the Vps75 dimer at either low protein or high salt concentration.

### Stoichiometry of co-purified Vps75-Rtt109 is 2:1

Having determined the stoichiometry of reconstituted Vps75–Rtt109 complexes, we sought to measure the stoichiometry of a co-expressed complex. Co-expression is more convenient and, in this case, produces higher yields than Rtt109 alone. Different expression and purification strategies can cause different stoichiometries, and our initial attempts using reported protocols resulted in heterogeneous samples (Tang et al, 2011). The first step in purification is Ni$^{2+}$-affinity chromatography via an N-terminal his tag on Rtt109 (Fig 2A). We found that a 5-ml HisTrap HP column (GE Healthcare) is saturated by complex from 6 liters of *Escherichia coli* culture. Increasing column volume or rerunning the flow-through did not

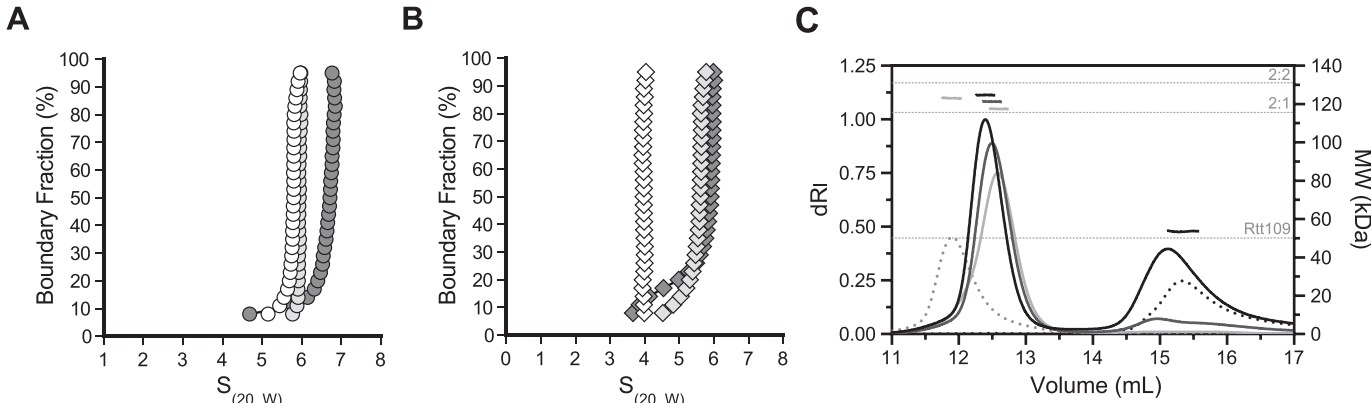

**Figure 1. Stoichiometry of Vps75–Rtt109 complexes.**
**(A, B)** van Holde–Weischet analysis of sedimentation velocity experiments of 7.5 μM Vps75 with 0, 0.5, and 1.0 equivalent (white, light grey, and dark grey, respectively) of Rtt109 at 150 mM NaCl (A, circles) or 300 mM NaCl (B, diamonds). Similar experiments for Rtt109 and H3-H4 are shown in Fig S1. **(C)** SEC-MALS of 10 μM Vps75 (dotted grey), 10 μM Rtt109 (dotted black), and 10 μM Vps75 with 5, 10, or 20 μM Rtt109 (light grey, dark grey, and black, respectively). Horizontal dashed lines indicate theoretical molecular weights of labeled complexes.

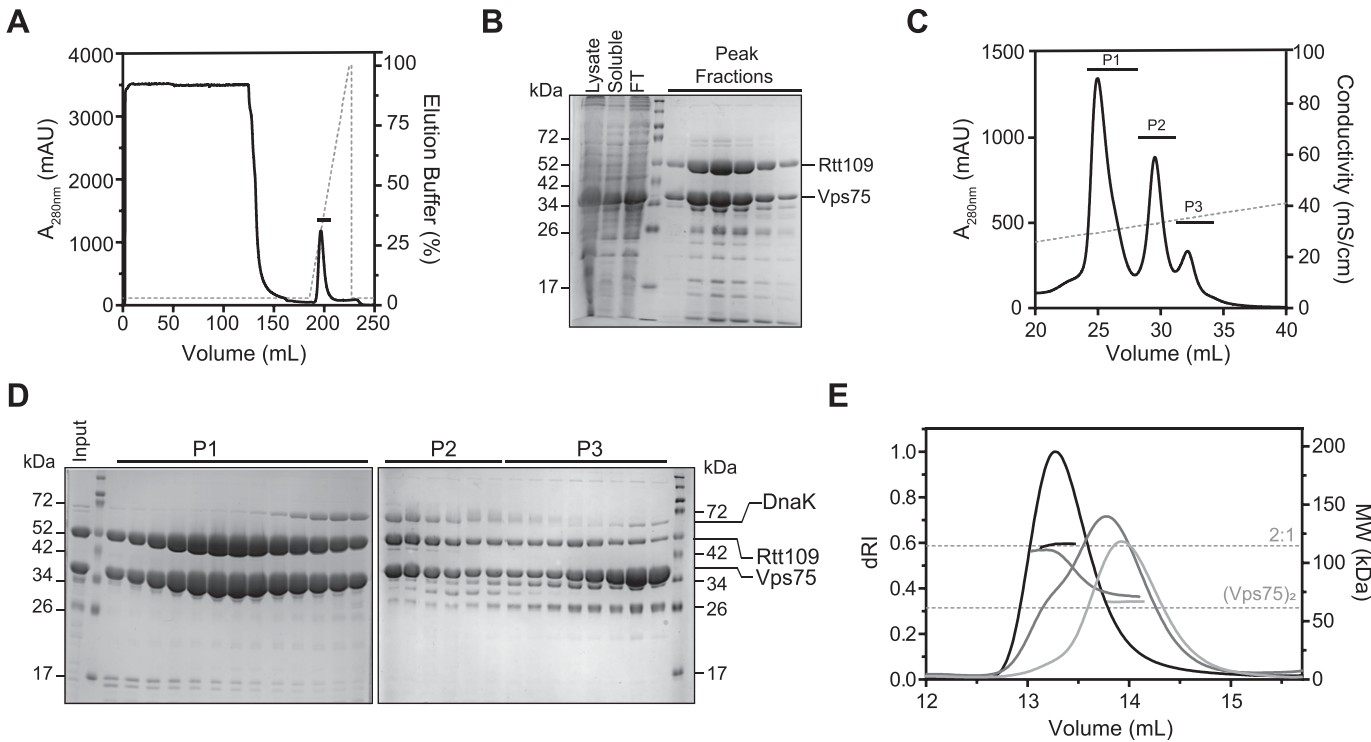

**Figure 2. Co-purification of Vps75–Rtt109.**
**(A, B, C, D)** Purification of co-expressed his–Rtt109 and Vps75 by Ni²⁺-affinity (A, B) and anion exchange (C, D) chromatography. FT is flow-through. **(E)** SEC-MALS of major peaks at 300 mM NaCl from anion exchange. P1, P2, and P3 are shown in black, dark grey, and light grey, respectively. Horizontal dashed lines indicate theoretical molecular weights of labeled complexes. Optimization of a MgATP wash is shown in Fig S2.

improve the yield. Notably, Vps75 was detected in the column flow-through, suggesting it was overexpressed compared with Rtt109, or that Rtt109 was depleted during purification (Fig 2B). The second step of purification is anion exchange, and three peaks containing Vps75 and Rtt109 eluted at 200–400 mM NaCl (Fig 2C). The most intense peak (P1) contained an ~70 kD contaminant, whereas the other two peaks (P2 and P3) had different relative amounts of Vps75 and Rtt109 (Fig 2D).

We improved the purification of co-expressed Vps75–Rtt109 by identifying and removing the ~70 kD contaminant. In-gel digestion followed by mass spectrometry showed it to be DnaK, an ATP-dependent, bacterial, folding chaperone. We recovered peptides for 75% of the DnaK sequence. DnaK can often be eliminated by the addition of MgATP (Rial & Ceccarelli, 2002). We tested several MgATP conditions and found the most efficient to be a 5 mM MgATP wash during Ni²⁺-affinity chromatography (Fig S2). This wash eliminated the unresolved shoulder on P1. We next ascertained the stoichiometry and homogeneity of the peaks using SEC-MALS (Fig 2E). P1 had a measured MW of 115.7 ± 2.5 kD, consistent with a 2:1 complex (theoretical MW of 114.8 kD). P2, on the other hand, was a mixture of at least two species with MWs near a 2:1 complex and unbound Vps75 dimer, whereas P3 is mostly an unbound Vps75 dimer. Clean separation of the 2:1 complex and unbound Vps75 dimer did not occur during gel filtration, so high stringency must be used when selecting anion exchange fractions if homogeneous preparations are desired. Ultimately, we have shown that our co-expressed complex does not retain the second, weakly bound copy of Rtt109 and has a stoichiometry of 2:1.

## H3-H4 displaces Rtt109 from a 2:2 Vps75–Rtt109 complex

We next wondered if the stoichiometry of Vps75–Rtt109 complexes influenced the binding of H3-H4 substrate. We again used sedimentation velocity analysis but titrated H3-H4 against Vps75–Rtt109 that was reconstituted at either 2:2 (Fig 3A, top) or 2:1 (Fig 3A, bottom). For the 2:2 complex, addition of 0.5 equivalents of H3-H4 caused an increase in sedimentation to 10 S with a tail trailing back to 3 S. Further increasing H3-H4 to 1.0 equivalent caused a subtle decrease in sedimentation of the 10 S species, and the appearance of an obvious noninteracting 3 S species. Comparison with controls (Fig 3A, faded) identified the noninteracting species to be Rtt109. Notably, a binary complex between Rtt109 and H3-H4 is not likely, as it was not found in solution (Fig S1B and C). Addition of H3-H4 thus caused Rtt109 to be released from the 2:2 Vps75–Rtt109 complex. Rtt109 remained in the complex at 0.5 equivalents of H3-H4, as a ternary Vps75–Rtt109–(H3-H4) complex was detected. This ternary complex sedimented at 10 S, which is greater than Vps75–(H3-H4) (9.3 S) or Vps75–Rtt109 (6.7 S) (Fig 3A, top).

One possibility is that Rtt109 is displaced by the (H3-H4)₂ tetramer, but not by an H3-H4 dimer. We tested this by performing similar titrations with an H3 dimerization mutant (H3_DM) (Fig 3B). The mutant contains three point mutations (C110E, L126A, and I130A) in the four-helix bundle responsible for tetramerization (Mattiroli et al, 2017). Unlike wild-type H3-H4, H3_DM-H4 sedimentation was not influenced by ionic strength, as it is exclusively dimeric (Fig S3). Addition of H3_DM-H4 to the 2:2 Vps75–Rtt109 complex caused a small

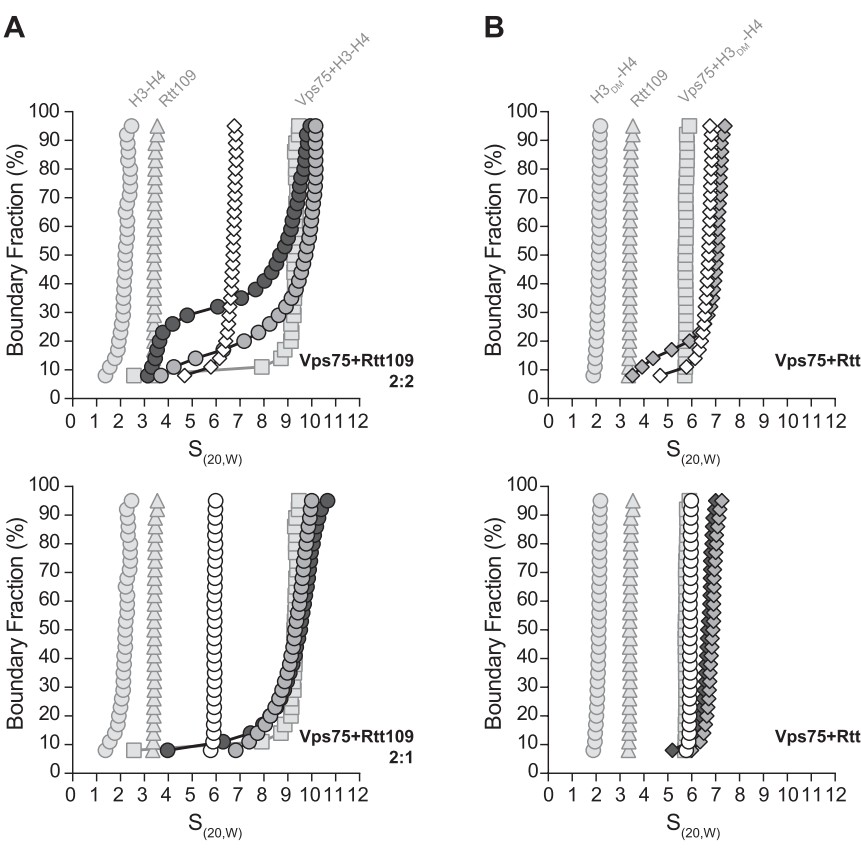

**Figure 3. Sedimentation velocity of the Vps75–Rtt109-(H3-H4) complex.**
**(A, B)** van Holde–Weischet analysis of sedimentation velocity experiments of complexes with Vps75, Rtt109, and H3-H4 (A) or H3$_{DM}$-H4 (B). Vps75 and Rtt109 were mixed 2:2 (top) or 2:1 (bottom). Vps75–Rtt109 is shown in white, and Vps75–Rtt109 and 0.5 or 1 equivalent of histones are shown in light grey and dark grey, respectively. For reference, sedimentation of H3-H4 or H3$_{DM}$-H4 (faded circles), Rtt109 (faded triangles), and a 2:2 mix of Vps75–(H3-H4) or Vps75–(H3$_{DM}$-H4) (faded squares) are also shown. Comparison of H3-H4 and H3$_{DM}$-H4 sedimentation is shown in Fig S3. All experiments were carried out at 150 mM NaCl.

increase in sedimentation and the appearance of a tail, comparable with that seen with wild-type H3-H4 (Fig 3B, top). This shows that one copy of Rtt109 is displaced by H3$_{DM}$-H4. A ternary Vps75–Rtt109-(H3$_{DM}$-H4) complex was detected at 7.1 S, which is greater than Vps75–(H3$_{DM}$-H4) (5.8 S), or Vps75–Rtt109 (6.7 S). Taken together, these data show that H3-H4 or H3$_{DM}$-H4 outcompete the weak binding of the second copy of Rtt109, revealing the stoichiometry of the ternary Vps75–Rtt109-(H3-H4) complex to be 2:1:1.

### 2:1:1 Vps75–Rtt109-(H3-H4) complex oligomerizes

Consistent results are obtained when titrating H3-H4 or H3$_{DM}$-H4 against a reconstituted 2:1 Vps75–Rtt109 complex (Fig 3A and B, bottom panels). Sedimentation shifts were like those obtained with the 2:2 complex, except that the unbound Rtt109 tail was absent. There was a small tail at 1.0 equivalent, likely attributed to free H3-H4 that is unable to compete with the tight binding of the first copy of Rtt109. The shape of the sedimentation plots at the perfect 2:2:1 ratio was also notable. For H3$_{DM}$-H4, the plot is vertical, suggesting a homogeneous 6.8 S complex (Fig 3B, bottom). The introduced H3 mutations prevented oligomerization of this complex, implying a stoichiometry of 2:2:1. By contrast, for wild-type H3-H4, the plot is sigmoidal, suggesting exchange between the 6.8 S complex and a 10 S complex (Fig 3A, bottom). This demonstrates that multiple copies of the 2:1:1 complex can dynamically associate to form a complex with a stoichiometry of 4:2:2. This association must be mediated by an H3-H3′ four-helix bundle that is mutated in H3$_{DM}$.

We next attempted to form the 2:1:1 complex at high concentration by adding H3-H4 or H3$_{DM}$-H4 to co-expressed Vps75–Rtt109. Initial attempts, however, were thwarted by precipitation with all samples being visually cloudy at or below 150 mM NaCl (Fig 4A). Such precipitation hindered our attempt to determine stoichiometry using SEC-MALS. As such, we prepared complexes at 6, 8, and 10 mg/ml and quantified solubility at various salt concentrations (Figs 4B and C and S4). We observe improved solubility up till 200 mM NaCl and saw only subtle differences between H3-H4 and H3$_{DM}$-H4. These observations suggest that precipitation is due to nonspecific ionic interactions, rather than the specific oligomerization that occurs via the H3-H3′ four-helix bundle. Based on these results, we elected to perform SEC-MALS at 300 mM NaCl to ensure solubility.

SEC-MALS of the co-expressed 2:1 Vps75–Rtt109 complex with H3-H4 or H3$_{DM}$-H4 gave results consistent with the sedimentation analysis. For H3-H4, mixtures eluted in a multimodal peak with a heterogeneous MW trace (Fig 5A). The MWs ranged from a theoretical 2:1:1 to 4:2:2, indicating oligomerization of a base 2:1:1 complex (Table S2). This heterogeneity was resolved when using H3$_{DM}$-H4. For H3$_{DM}$-H4, mixtures eluted in a single dominant peak with a homogeneous MW trace (Fig 5B). In both cases, the MW was close to a theoretical 2:1:1 complex (Table S2). Gel filtration of large-scale purifications further confirmed inclusion of the four proteins and oligomerization of the complex with wild-type H3-H4 (Fig 5C). The Vps75–Rtt109-(H3-H4) complex is thus composed of one or two copies of a stable 2:1:1 sub-complex. The two copies come together through the formation of an H3-H3′ four-helix bundle.

**A**

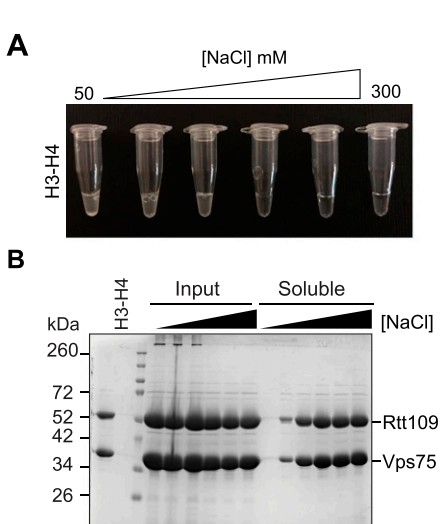

**B**

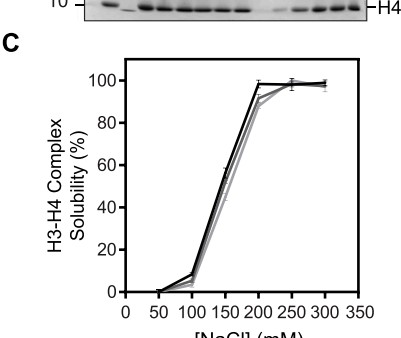

**C**

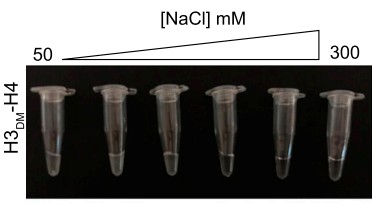

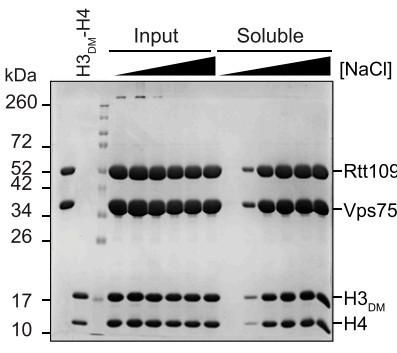

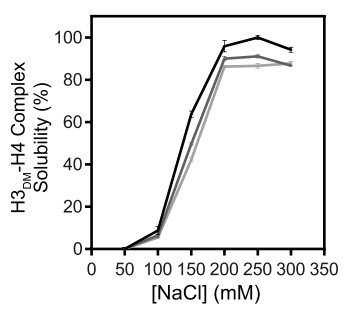

**Figure 4. Optimizing solubility of Vps75–Rtt109–(H3-H4) complexes.**

Solubility of complexes containing H3-H4 (left) or H3$_{DM}$-H4 (right) at 6 mg/ml was assayed as a function of NaCl concentration. **(A)** Image showing visual turbidity. **(B)** SDS–PAGE of samples before (Input) and after (Soluble) centrifugation. **(C)** Percent solubility of complexes at 6, 8, and 10 mg/ml based on UV spectroscopy before and after centrifugation (black, grey, and light grey, respectively). Complete data for 8 and 10 mg/ml are shown in Fig S4.

# Discussion

We have investigated the stoichiometry of Vps75–Rtt109 using both reconstituted and co-purified complexes. We were prompted to do this as multiple crystal structures were reported showing that Vps75 can bind to Rtt109 at 2:2 or 2:1 (Su et al, 2011; Tang et al, 2011). We find that both the 2:2 and 2:1 complex can be reconstituted, depending on protein concentration and buffer ionic strength. The second copy of Rtt109 is easily lost, indicating a weaker binding affinity. This is consistent with previous analytical ultracentrifugation data, NMR, and analytical gel filtration data where a 2:2 complex is observed, albeit less occupied (Park et al, 2008; Tang et al, 2008; Danilenko et al, 2019). We further show that the 2:1 Vps75–Rtt109 complex does not oligomerize. This shows that Rtt109 binding to the Vps75 dimer interferes with Vps75 tetramer formation (Hammond et al, 2016). The Vps75 dimer has loose twofold symmetry as it is held together by head-to-tail packing of two helices, one from each chain, in a coiled coil (Tang et al, 2008). It is not immediately apparent what breaks this symmetry to limit the affinity of the second copy of Rtt109. Given that a 2:2 complex can be populated sufficiently to obtain a crystal structure (Tang et al, 2011), asymmetry is not likely driven by binding of Rtt109 or any associated steric hindrance. Rather, we suggest that the Vps75 dimer itself can switch between symmetric

and asymmetric conformations. This switch may be modulated by extrinsic factors such as ionic strength or the binding of another ligand such as H3-H4.

The 2:1 Vps75–Rtt109 complex is also the most relevant for H3-H4 acetylation. We demonstrate this by showing that addition of H3-H4 to the 2:2 complex causes at least one copy of Rtt109 to be evicted. Our experiments are performed in the micromolar regime, and H3-H4 replaces the weakly bound Rtt109 at a stoichiometric ratio. This shows that the complex has a greater affinity for H3-H4 than the weakly bound Rtt109. Importantly, H3-H4 does not evict all copies of Rtt109 from the complex and a ternary Vps75–Rtt109–(H3-H4) complex forms. Given the comparable affinities between Vps75 and either H3-H4 or Rtt109 (Tsubota et al, 2008; Kolonko et al, 2010), complete Rtt109 eviction would require a vast excess of H3-H4. It would also require overlap (or at least an allosteric connection) between the Rtt109 and H3-H4 binding sites. It is possible that the distribution of Vps75 between different complexes with Rtt109, H3-H4, or both Rtt109 and H3-H4 is cell cycle dependent. Expression of both H3-H4 and Rtt109 peaks during S phase, whereas Vps75 expression remains relatively constant (Osley, 1991; Driscoll et al, 2007; Selth et al, 2009). Given the relevance of the 2:1 complex, we report an optimized protocol for its purification. This protocol is invaluable for future enzymatic and structural studies where homogeneous samples are required.

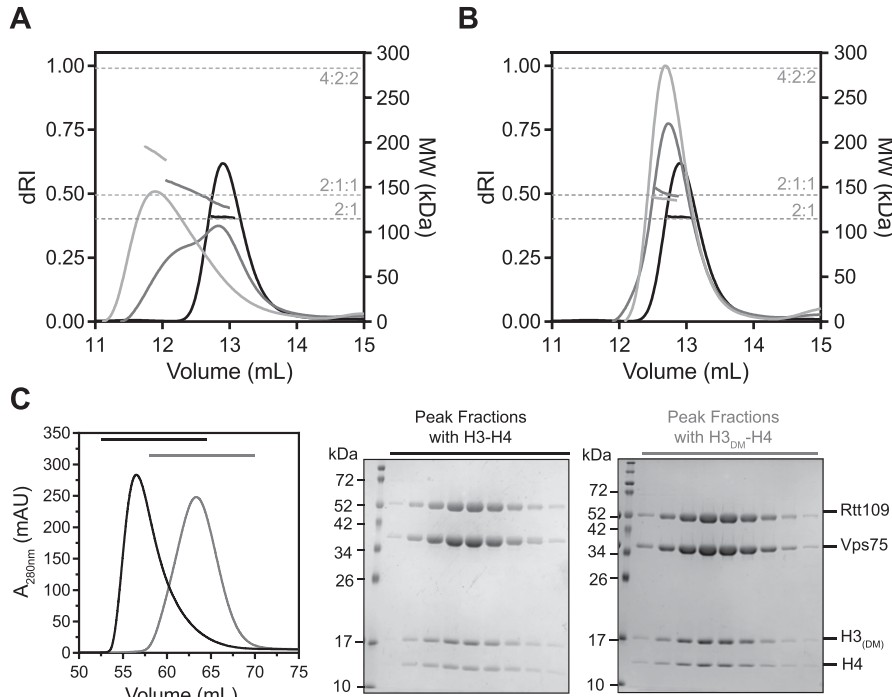

**Figure 5. Stoichiometry and large-scale preparation of Vps75–Rtt109–(H3-H4).**
**(A, B)** SEC-MALS of 20 $\mu$M 2:1 Vps75–Rtt109 (black) with 20 or 40 $\mu$M (dark grey and light grey, respectively) H3-H4 (A) or H3$_{DM}$-H4 (B). Horizontal dashed lines indicate theoretical molecular weights of labeled complexes. Molecular weights are listed in Table S2. **(C)** Gel filtration of large-scale purifications of Vps75–Rtt109–(H3-H4) (black) and Vps75–Rtt109–(H3$_{DM}$-H4) (grey). SDS–PAGE of peak fractions.

Finally, we have determined the stoichiometry and arrangement of the ternary Vps75–Rtt109–(H3-H4) complex. We could do this as we defined the optimal conditions for large-scale reconstitution of an H3-H4–bound Vps75–Rtt109 complex. We show that the 2:1 Vps75–Rtt109 complex can accommodate a single copy of an H3-H4 dimer or an (H3-H4)$_2$ tetramer. The dimer-bound complex is homogeneous, whereas the tetramer-bound complex contains several species in dynamic equilibrium. Because the H3 contained mutations that prevent an H3-H3′ four-helix bundle, this equilibrium is attributed to the complex-forming dimers. The ternary complex of Vps75–Rtt109–(H3-H4) is thus composed of a 2:1:1 subcomplex that can dimerize to form a 4:2:2 complex. This dimerization is mediated by a four-helix bundle involving the two copies of H3. Our H3 mutant may also mimic the effects of another chaperone, Asf1, which prevents H3-H4 forming a tetramer (English et al, 2005).

Our study comes after work by Danilenko and colleagues characterizing Rtt109–(H3-H4) with both chaperones Vps75 and Asf1 (Danilenko et al, 2019). We consistently report that the basic unit of Vps75–Rtt109–(H3-H4) has a stoichiometry of 2:1:1. It remains to be seen if the in vivo complex contains Vps75, Asf1, or both chaperones, and all possible complexes warrant study. Our focus on the complex containing only Vps75, which has been detected in vivo, has allowed us to make several additional conclusions (Fillingham et al, 2008). By performing titrations at various ionic strengths, we show that there are two unequal binding sites for Rtt109 in the Vps75 dimer. The weaker of these sites is easily displaced by H3-H4, regardless of whether H3-H4 is a dimer or a tetramer. This is significant as it directly shows that the second copy of Rtt109 is not relevant for the Vps75-driven acetylation of either an H3-H4 dimer or tetramer. We show that the complex between Vps75, Rtt109, and

H3-H4 exists in dynamic equilibrium between a 2:1:1 complex and a 4:2:2 complex. This equilibrium is driven by contacts between two copies of H3 that mediate the formation of a four-helix bundle. Four-helix bundles are common interfaces between histone dimers (Arents et al, 1991; Luger et al, 1997). This is a new insight into the arrangement of the complex that has repercussions for ongoing structural and drug discovery programs focused on this complex. These conclusions, as well as our optimized protocols for purification of relevant complexes, pave the way for mechanistic and high-resolution structural studies.

# Materials and Methods

### Protein expression and purification

*Saccharomyces cerevisiae* Vps75 and Rtt109 were individually expressed and purified as previously described (Park et al, 2008). The Vps75–Rtt109 complex was co-expressed and purified as described previously (Tang et al, 2011). Both genes were expressed from a pRSFDuet-1 vector in Rosetta (DE3) pLysS cells grown in 2xTY. Rtt109 was in multiple cloning site 1, and Vps75 was in multiple cloning site 2. Expression was induced at an OD$_{590nm}$ of 0.7–0.8 with 0.5 mM IPTG and the cells left at 37°C overnight. 6 liters of culture was lysed by three passes through a homogenizer at 18,000 $\psi$. The lysis buffer contained 20 mM Tris–HCl pH 7.5, 500 mM NaCl, 5 mM BME, and 0.2 mM AEBSF with additional protease inhibitors (aprotinin [5 $\mu$g/ml], leupeptin [1 $\mu$g/ml], and pepstatin [1 $\mu$g/ml]). The soluble fraction was loaded onto a 5-ml HisTrap column (GE Healthcare) that was pre-equilibrated in 20 mM Tris–HCl pH 7.5, 300 mM NaCl, 5 mM BME, and 0.2 mM AEBSF with 15 mM imidazole. After

washing to baseline, the complex was eluted using a 15–500 mM imidazole gradient over 8 column volumes (CVs). Before elution, the column was washed with 5 mM MgATP as described in Fig S2. Peak fractions from the HisTrap elution were combined and diluted 1 in 2 with 20 mM Tris–HCl pH 7.5, 5 mM BME, 0.2 mM AEBSF, and 0.5 mM EDTA. The diluted sample was loaded onto a MonoQ 10/100 GL column (GE Healthcare) pre-equilibrated in 20 mM Tris–HCl pH 7.5, 150 mM NaCl, 5 mM BME, 0.2 mM AEBSF, and 0.5 mM EDTA. After washing to baseline, the complex was eluted using a 150–1,000 mM NaCl gradient over 12 CVs. Purification was monitored using 15% SDS–PAGE stained with Coomassie Blue. When required, gels were quantified using Image Lab 1.1.0.04. The total intensity of each lane was set to 100%. In-gel digestion and mass spectrometry were performed using standard protocols by the Institutional Mass Spectrometry Core Laboratory at the University of Texas Health Science Center, San Antonio.

Unfolded *Xenopus laevis* wild-type H3, mutant H3$_{DM}$ (C110E, L126A, and I130A), and H4 were purchased from the Histone Source at Colorado State University. They were refolded using standard protocols (Dyer et al, 2003).

### Sedimentation velocity analysis

Proteins were dialyzed into 20 mM Tris pH 7.5, 150 mM or 300 mM NaCl, and 0.5 mM TCEP. Mixtures were formed relative to 7.5 $\mu$M Vps75 (calculated with monomer extinction coefficient) (Figs 1 and 3), 7.5 $\mu$M Rtt109 (Fig S1), and 22.5 $\mu$M H3-H4 or H3$_{DM}$-H4 (calculated with the H3-H4 dimer extinction coefficient) (Fig S3). Sedimentation velocity experiments were performed using a Beckman Coulter Optima XL-A or XL-I analytical ultracentrifuge. Samples were placed in standard epon 2-channel centerpiece cells. Sedimentation was then monitored using either absorbance or intensity modes at 229 or 280 nm at 20°C. Data were collected using speeds of 30,000, 35,000, 45,000, or 50,000 rpm with either an An60Ti or An50Ti rotor. Partial specific volumes were determined using UltraScan3 version 2.0. Time invariant and radial invariant noise was subtracted from the sedimentation velocity data by two-dimensional spectrum analysis, followed by genetic algorithm refinement and Monte Carlo analysis (Demeler & Brookes, 2008). Sedimentation coefficient distributions G(s) were obtained with enhanced van Holde–Weischet analysis (Demeler et al, 1997). Calculations were performed on the UltraScan LIMS cluster at the Bioinformatics Core Facility, University of Texas Health Science Center, San Antonio; and the Lonestar cluster at the Texas Advanced Computing Center, supported by NSF Teragrid Grant #MCB070038.

### SEC-MALS

For analysis of Vps75–Rtt109 only (Fig 1C), proteins were dialyzed into 20 mM Tris pH 7.5, 150 mM NaCl, and 1 mM TCEP. Mixtures were formed relative to 10 $\mu$M Vps75. Rtt109 alone was also prepared at 10 $\mu$M. 100 $\mu$l of each sample was injected over a Superdex 200 10/300 GL column using an ÄKTA purifier HPLC system (GE Healthcare) at 0.3 ml/min. This system was directly connected to a Dawn Heleos II multi-angle light scattering instrument and a refractive index detector (Wyatt Technologies). Data were analyzed with ASTRA 5.

For analysis of Vps75–Rtt109 anion exchange peaks (Fig 2E) and Vps75–Rtt109–(H3-H4) complexes (Fig 5A), samples were dialyzed into to 20 mM Tris–HCl pH 7.5, 300 mM NaCl, and 0.5 mM TCEP. Samples were concentrated to 20 $\mu$M or mixtures formed relative to 20 $\mu$M Vps75. 100 $\mu$l of each sample was injected over a Superdex 200 Increase 10/300 GL column using an ÄKTA pure HPLC system (GE Healthcare) at 0.75 ml/min. This system was directly connected to a miniDAWN TREOS and Optilab T-rEX refractive index detectors (Wyatt Technologies). Data were analyzed with ASTRA 7. Detailed protocols are published (Sarkar et al, 2020).

### Solubility analysis

Concentrated stocks of 2:1 Vps75–Rtt109, H3-H4, and H3$_{DM}$-H4 were prepared in 20 mM Tris–HCl pH 7.5, 300 mM NaCl, and 0.5 mM TCEP. Vps75–Rtt109 and histones were then combined 1:1 and diluted to 6, 8, or 10 mg/ml. The dilution buffer was adjusted to give a final NaCl concentration of 50–300 mM in 50 mM increments. Aliquots were taken before (input) and after (soluble) 2-min centrifugation at 8,800$g$ and analyzed by 15% SDS–PAGE stained with Coomassie Blue. Triplicate UV spectra (190–850 nm) were also measured before and after centrifugation using a NanoDrop One/One$^C$ UV-Vis Spectrophotometer. Mean data were plotted using GraphPad Prism 8 showing 95% confidence intervals.

### *Large-scale gel filtration*

Vps75–Rtt109-(H3-H4) and Vps75–Rtt109–(H3$_{DM}$-H4) were formed by mixing co-expressed Vps75–Rtt109 and histone 1:1. Samples were then injected over a Superdex 200 16/60 prep grade column (GE Healthcare) pre-equilibrated in 20 mM Tris–HCl pH7.5, 300 mM NaCl, and 0.5 mM TCEP. Peak fractions were run on 15% SDS–PAGE and visualized with Coomassie Blue.

## Supplementary Information

## Acknowledgements

We are grateful to Borries Demeler (University of Lethbridge) for advice on the analysis of analytical ultracentrifugation data, Andrew Andrews (Fox Chase Cancer Center) for providing the Vps75–Rtt109 co-expression plasmid, and Karolin Luger (Howard Hughes Medical Institute and University of Colorado) for early support of the project. We thank Lokeshwar Bhenderu for assistance with purifications and other members of the D'Arcy group for comments on the manuscript. Funding: This work was supported by grants to S D'Arcy from the National Institutes of Health R35 GM1337551 and the University of Texas at Dallas.

### Author Contributions

N Akhavantabib: conceptualization, data curation, formal analysis, investigation, and writing—original draft.
DD Krzizike: conceptualization, data curation, formal analysis, investigation, and methodology.

V Neumann: investigation.

S D'Arcy: conceptualization, data curation, formal analysis, supervision, funding acquisition, investigation, and writing—review and editing.

## Conflict of Interest Statement

The authors declare they have no conflicts of interest with the contents of this article.

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
