## [Reviewer comments · Life Science Alliance]

Life Science Alliance

Stoichiometry of Rtt109 complexes with Vps75 and histones H3-H4

Noushin Akhavantabib, Daniel Krzizike, Victoria Neumann, and Sheena D'Arcy
DOI: <https://doi.org/10.26508/lsa.202000771>

Corresponding author(s): Sheena D'Arcy, The University of Texas at Dallas

Review Timeline:

Submission Date:	2020-05-11
Editorial Decision:	2020-06-30
Appeal Requested:	2020-07-20
Editorial Decision:	2020-07-28
Revision Received:	2020-08-13
Editorial Decision:	2020-08-14
Revision Received:	2020-08-31
Accepted:	2020-09-01

Scientific Editor: Shachi Bhatt

Transaction Report:

June 30, 2020

Re: Life Science Alliance manuscript #LSA-2020-00771

Dr. Sheena D'Arcy
The University of Texas at Dallas
Chemistry and Biochemistry
800 West Campbell Rd
Richardson, TX 75080

Dear Dr. D'Arcy,

Thank you for submitting your manuscript entitled "Absolute stoichiometry of Rtt109 complexes with Vps75 and histones H3-H4" to Life Science Alliance. The manuscript has now been seen by expert reviewers, whose reports are appended below. Unfortunately, after an assessment of the reviewer feedback, our editorial decision is against publication in Life Science Alliance.

While the reviewers agree that your work defining the stoichiometries of various Rtt109/Vps75/H3/H4 complexes is well-performed overall, both Reviewers 1 and 3 are concerned that the insights to be gained with respect to biological function or preparation for future high-resolution structural studies are small, and that these insights have been somewhat pre-empted by previous work (Danilenko, Nat Comm 2019).

Given these concerns, we are afraid we do not have the level of reviewer support that we would need to proceed further with the paper. We are thus returning your manuscript to you with the message that we cannot publish it here.

We are sorry our decision is not more positive, but hope that you find the reviews constructive. Of course, this decision does not imply any lack of interest in your work and we look forward to future submissions from your lab.

Thank you for your interest in Life Science Alliance.

Sincerely,

Reilly Lorenz
Editorial Office Life Science Alliance
Meyerhofstr. 1
69117 Heidelberg, Germany
t +49 6221 8891 414
e contact@life-science-alliance.org
www.life-science-alliance.org

Reviewer #1 (Comments to the Authors (Required)):

In the present manuscript entitled "Absolute stoichiometry of Rtt109 complexes with Vps75 and histones H3-H4" by Akhavantabib et al., the authors thoroughly analyse the complex composition and binding stoichiometries of the Vps75-Rtt109-H3/4 complex. In addition, the authors established a protocol for large-scale preparation of the H3/4-bound Vps75-Rtt109 complex for in depth future studies. While the study is technically sound (as far as I can judge as I am no expert in sedimentation velocity), the biological insight is very limited. As it stands and without any further insight into the biology of Vps75-Rtt109, the study is very preliminary and would require additional work to provide an advancement for the field in terms of mechanistic insight. Given the recent publication by the Carlomagno lab (Danilenko et al., Nat Comm 2019) that describes a 2:1 Vps75-Rtt109 complex (deemed in this present manuscript as most relevant - see below) in complex with Asf1, the additional insight into the function of Vps75-Rtt109 is small.

- In the discussion the authors claim that the 2:1 Vps-Rtt109 complex is the most relevant for H3-H4 acetylation. In order to prove this, enzymatic assays under the respective conditions are required.
- The 2:2 Vps75-Rtt109 complex appears to be very weak and it is not clear if this occurs in vivo. Thus, it would be very beneficial to understand which complex composition occurs in vivo
- It would be helpful to include SDS-PAGE analysis for the size exclusion experiments (particularly for 1C) to help visualize the composition of the complex
- The manuscript is sometimes difficult to follow as it quickly switches between the different complex stoichiometries - it would be good to include little illustrations in the figures
- In the introduction, the authors refer to H3K56ac as important in DNA repair. While this is obviously correct, H3K56ac has been implicated in many other chromatin-templated processes, including transcription and DNA replication

Reviewer #2 (Comments to the Authors (Required)):

Akhavantabib et al. describe the stoichiometry of complexes of the HAT/PAT Rtt109 with Vps75 and one of their substrates, histones H3-H4, under a variety of conditions, and also the strategies they developed to purify homogenous complexes at a large scale. The work uses stringent biophysical methods combined with appropriate mutations in the proteins to produce high quality data resulting in high confidence in the results and conclusions. Previous results from crystallographic studies raised some confusion about the stoichiometry of these complexes, but these authors resolve these issues satisfactorily by demonstrating the conditions under which each complex can be reproducibly obtained. While the studies presented here do not reveal new insights into the functions of Rtt109 or its cofactors, they provide a solid basis for future structural and enzymological analyses by sorting out some confusion in the literature and providing a sound basis for obtaining high quality reagents. The manuscript is well written and the data are clearly presented, with minor exceptions noted below.

Page 4: "if wanting homogenous preparations" should be "if homogenous preparations are desired"

Throughout: the manuscript shifts back and forth between the more standard past tense and the less standard present tense when describing experimental results. One convention should be chosen and adhered to throughout.

Reviewer #3 (Comments to the Authors (Required)):

1. General comments.

Vps75 is known to form complexes with Rtt109 and H3-H4 but the relative stoichiometries of these different complexes remains unclear. Sedimentation velocity and SEC-MALS experiments show that a VPS75 dimer can bind a single or two copies of RTT109 as a function of concentration and ionic strength. A 2:1 stoichiometry predominates in agreement with a structural model derived from NMR data (Danileno et al 2019). Such a complex can engage a H3-H4 dimer or tetramer but the biologically relevant complex remains unclear.

While such studies are important to enable subsequent high-resolution structural studies, the conclusions drawn are somewhat pre-empted by previously published structural data on the VPS75-Rtt109-H3-H4-Asf1 complex (Danileno et al 2019). The experimental evidence and technical quality of the is solid and confirm the makeup of the complex seen by Danileno et al. Beyond the confirmatory nature, these data only provide however only a limited conceptual advance to what already has been published.

2. Major comments:

1. As the authors' data indicate that Vps75 and RTT109 form a dynamic equilibrium at 2:1 or 2:2, depending on conditions, it is not clear why they insist that they have determined the 'absolute stoichiometry'. On possible scenario, also mentioned by the authors, is that different stoichiometries exist also as a function of H3-H4 substrate binding.

2. Discussion: the authors indicate that '...our H3 mutant may also be a surrogate for another chaperone...'. As such a mutant does not exist in cells, it can not be a surrogate for a chaperone.

3. While it is important to understand the stoichiometry of such complexes, to understand the underlying mechanisms, the motivation for

3. Minor comments:

1. Title: As the complexes form a dynamic equilibrium revise the title 'Absolute stoichiometry...' maybe just 'Stoichiometry of Rtt109 complexes with Vps75 and histones H3-H4' as in the running title.

2. Table S1: Why not also incorporate results from the AUC such that the reader has an overview over samples analysed and results obtained using the different techniques?

Re: Reconsider manuscript #LSA-2020-00771

I politely request that you revisit the decision to reject our manuscript entitled “Absolute stoichiometry of Rtt109 complexes with Vps75 and histone H3-H4”. We strongly believe it is suitable for publication in your journal.

Our manuscript was rejected because of overlap with a study by Danilenko et al. (Nat Comm 2019). Two reviewers felt that this overlap limited the novelty of the biological insights gained. We were of course aware of this study and my student (the first author) was disappointed to be scooped on some of our major findings. Our work was mostly completed when Danilenko et al. was published and our study is truly an independent verification.

In response to this situation, I refer to your editorial policies stating that you are committed to reporting “*confirmatory*” data and that you “*encourage replicating work if the results are either highly important... or if the work extends the scope of previous work*”. Our manuscript includes data that confirm and extend the findings in Danilenko et al. Key points are as follows.

1. We show that the Vps75 dimer has a strong Rtt109 binding site and a weak Rtt109 binding site, and we assess the impact of mixture stoichiometry and ionic strength. The latter is particularly important as Vps75 oligomerizes in a salt-dependent manner. By addressing these two additional variables we confirm and build upon the similar result reported by Danilenko et al.
2. We directly show that H3-H4 will kick-out the weakly-bound or ‘second’ copy of Rtt109. The biological significance of this is high as it directly shows that the second copy of Rtt109 is not relevant for the Vps75-driven acetylation of H3-H4. Further, we show that Rtt109 will be kicked-out by either a H3-H4 dimer or a H3-H4 tetramer. The biological significance of this is also high as it highlights that both forms of H3-H4 are potential substrates of Rtt109. In contrast, Danilenko et al. characterizes the complex with a different substrate, the complex between Asf1 and H3-H4. Our conclusions are thus distinct from those in Danilenko et al. Our direct competition experiments are particularly notable and novel.

3. Danilenko et al. focus on the complex containing both chaperones, Vps75 and Asf1. Currently there is no evidence that this two-chaperone complex occurs and is the major complex in vivo. We focus on the complex containing only Vps75. This complex has been detected in vivo and has been shown to have distinct activity from other complexes containing Asf1. For this reason, studying both the single and double-chaperone complexes is warranted.
4. We show that the complex between Vps75, Rtt109 and H3-H4 exists in dynamic equilibrium between a 2:1:1 complex and a 4:2:2 complex. This equilibrium is driven by contacts between two copies of H3 that mediate the formation of a four-helix bundle. Four-helix bundles are common interfaces between histone dimers. This is a new insight into the arrangement of the complex that has repercussions for ongoing structural and drug-discovery programs focused on this complex.
5. We optimize and clearly define protocols for the purification of the Vps75, Rtt109, H3-H4 complex. This is important and should alter how other laboratories prepare their complexes to ensure maximum solubility and precise stoichiometry. Again, this is essential information for structural and translational programs looking at this complex.

In reading the reviewers' comments it becomes apparent that we did not present our data in a way that best highlighted the biological significance of our findings. We focused too much on the technical aspects and the implications of our findings to future structural studies. We of course would be willing to alter the text if it please the reviewers and editors. As an early career researcher, I am always learning how to best represent our data.

Finally, I wish to reiterate that our manuscript is of interest to those in the chromatin field, the biophysics community, and a general biochemical audience. It is high-quality work that confirms and extends the findings in Danilenko et al. Both reviewers 2 and 3 commented on the high quality of our work and the confidence they had in our conclusions. I hope that you agree that such high-quality work belongs in your journal.

Thank you for considering this request.

MS: LSA-2020-00771

Dr. Sheena D'Arcy
The University of Texas at Dallas
Chemistry and Biochemistry
800 West Campbell Rd
Richardson, TX 75080

Dear Dr. D'Arcy,

Your manuscript entitled "Absolute stoichiometry of Rtt109 complexes with Vps75 and histones H3-H4" has now been reconsidered, and we would reconsider a revised manuscript. As you suggested, please highlight much more clearly the biological significance of your data and compare and contrast your work explicitly and clearly with the previous work of Danilenko et al. We empathize with your predicament and given the technical quality of the work and the possibility of clarifying these points (and our policy of considering confirmatory data), we are happy to reverse the rejection. In the revised manuscript, we would ask that you also respond to all the reviewers' comments (most of which are in regard to interpretation and not experimental revision).

Yours sincerely,

Reilly Lorenz
Editorial Office Life Science Alliance
Meyerhofstr. 1
69117 Heidelberg, Germany
t +49 6221 8891 414
e contact@life-science-alliance.org
www.life-science-alliance.org

Response to Reviewers

We thank all the reviewers for their comments on the manuscript.

Reviewer #1 (Comments to the Authors (Required)):

In the present manuscript entitled "Absolute stoichiometry of Rtt109 complexes with Vps75 and histones H3-H4" by Akhavantabib et al., the authors thoroughly analyse the complex composition and binding stoichiometries of the Vps75-Rtt109-H3/4 complex. In addition, the authors established a protocol for large-scale preparation of the H3/4-bound Vps75-Rtt109 complex for in depth future studies. While the study is technically sound (as far as I can judge as I am no expert in sedimentation velocity), the biological insight is very limited. As it stands and without any further insight into the biology of Vps75-Rtt109, the study is very preliminary and would require additional work to provide an advancement for the field in terms of mechanistic insight. Given the recent publication by the Carlomagno lab (Danilenko et al., Nat Comm 2019) that describes a 2:1 Vps75-Rtt109 complex (deemed in this present manuscript as most relevant - see below) in complex with Asf1, the additional insight into the function of Vps75-Rtt109 is small.

We have added a new concluding paragraph to address the consistencies with Danilenko et al. and highlight our additional biological insights.

- In the discussion the authors claim that the 2:1 Vps-Rtt109 complex is the most relevant for H3-H4 acetylation. In order to prove this, enzymatic assays under the respective conditions are required.

The respective conditions would be Vps75-Rtt109 at 2:2 or 2:1. It is not essential to compare enzyme activity with these two complexes as we show that the substrate (H3-H4) competes the second copy of Rtt109 off Vps75. This is shown in Figure 3A, top panel. In other words, the 2:2 complex is likely not relevant for acetylating H3-H4 as the presence of H3-H4 makes it a 2:1 complex.

- The 2:2 Vps75-Rtt109 complex appears to be very weak and it is not clear if this occurs in vivo. Thus, it would be very beneficial to understand which complex composition occurs in vivo

We agree that it would be great to know the complex composition in vivo, but it is outside the scope of this work. We hope that our in vitro work will guide experiments aimed at ascertaining the most relevant in vivo complex.

- It would be helpful to include SDS-PAGE analysis for the size exclusion experiments (particularly for 1C) to help visualize the composition of the complex

Figure 1C and other SEC-MALS experiments were done at concentrations too low to clearly visualize on an SDS-PAGE. The composition of our final complexes however is visualized by SDS-PAGE in Figure 5C. In this figure, bands for Rtt109, Vps75, H3 and H4 are evident.

- The manuscript is sometimes difficult to follow as it quickly switches between the different complex stoichiometries - it would be good to include little illustrations in the figures

We did attempt to add cartoons but found it that most figures did not contain enough space to nicely incorporate them. However, we now include Table S2 that outlines the sedimentation velocity experiments and we hope it allows the reader to understand our ratios more easily.

- In the introduction, the authors refer to H3K56ac as important in DNA repair. While this is obviously correct, H3K56ac has been implicated in many other chromatin-templated processes, including transcription and DNA replication

Thank you for this note. We have adjusted the second paragraph of the introduction accordingly.

Reviewer #2 (Comments to the Authors (Required)):

Akhavantabib et al. describe the stoichiometry of complexes of the HAT/PAT Rtt109 with Vps75 and one of their substrates, histones H3-H4, under a variety of conditions, and also the strategies they developed to purify homogenous complexes at a large scale. The work uses stringent biophysical methods combined with appropriate mutations in the proteins to produce high quality data resulting in high confidence in the results and conclusions. Previous results from crystallographic studies raised some confusion about the stoichiometry of these complexes, but these authors resolve these issues satisfactorily by demonstrating the conditions under which each complex can be reproducibly obtained. While the studies presented here do not reveal new insights into the functions of Rtt109 or its cofactors, they provide a solid basis for future structural and enzymological analyses by sorting out some confusion in the literature and providing a sound basis for obtaining high quality reagents. The manuscript is well written and the data are clearly presented, with minor exceptions noted below.

Page 4: "if wanting homogenous preparations" should be "if homogenous preparations are desired"

The text has been revised to "if homogenous preparations are desired".

Throughout: the manuscript shifts back and forth between the more standard past tense and the less standard present tense when describing experimental results. One convention should be chosen and adhered to throughout.

Thank you for pointing this out, we have changed to the more standard past tense throughout.

Reviewer #3 (Comments to the Authors (Required)):

1. General comments.

Vps75 is known to form complexes with Rtt109 and H3-H4 but the relative stoichiometries of

these different complexes remains unclear. Sedimentation velocity and SEC-MALS experiments show that a VPS75 dimer can bind a single or two copies of RTT109 as a function of concentration and ionic strength. A 2:1 stoichiometry predominates in agreement with a structural model derived from NMR data (Danileno et al 2019). Such a complex can engage a H3-H4 dimer or tetramer but the biologically relevant complex remains unclear.

While such studies are important to enable subsequent high-resolution structural studies, the conclusions drawn are somewhat pre-empted by previously published structural data on the VPS75-Rtt109-H3-H4-Asf1 complex (Danileno et al 2019). The experimental evidence and technical quality of the is solid and confirm the makeup of the complex seen by Danileno et al. Beyond the confirmatory nature, these data only provide however only a limited conceptual advance to what already has been published.

We have added a new concluding paragraph to address the consistencies with Danilenko et al. and highlight our additional biological insights.

2. Major comments:

1. As the authors' data indicate that Vps75 and RTT109 form a dynamic equilibrium at 2:1 or 2:2, depending on conditions, it is not clear why they insist that they have determined the 'absolute stoichiometry'. On possible scenario, also mentioned by the authors, is that different stoichiometries exist also as a function of H3-H4 substrate binding.

We intended the word 'absolute' to differentiate between understanding the ratio of proteins in the complex from the actual oligomeric state, such as 2:1 or 4:2. We can see how this is confusing and have modified the text and title accordingly.

2. Discussion: the authors indicate that '...our H3 mutant may also be a surrogate for another chaperone...'. As such a mutant does not exist in cells, it can not be a surrogate for a chaperone.

We have removed the term surrogate.

3. While it is important to understand the stoichiometry of such complexes, to understand the underlying mechanisms, the motivation for

3. Minor comments:

1. Title: As the complexes form a dynamic equilibrium revise the title 'Absolute stoichiometry...' maybe just 'Stoichiometry of Rtt109 complexes with Vps75 and histones H3-H4' as in the running title.

Title has been modified.

2. Table S1: Why not also incorporate results from the AUC such that the reader has an overview over samples analysed and results obtained using the different techniques?

We have added Table S2 that summarizes the AUC experiments performed.

August 14, 2020

RE: Life Science Alliance Manuscript #LSA-2020-00771R-A

Dr. Sheena D'Arcy
The University of Texas at Dallas
Chemistry and Biochemistry
800 West Campbell Rd
Richardson, TX 75080

Dear Dr. D'Arcy,

Thank you for submitting your revised manuscript entitled "Stoichiometry of Rtt109 complexes with Vps75 and histones H3-H4". We would be happy to publish your paper in Life Science Alliance pending final revisions necessary to meet our formatting guidelines.

Please make the following edits in your revised manuscript,

- please add the Author Contributions to the main manuscript text
- please add your abstract into our system
- please upload both your main and supplementary figures as separate files
- please add a separate section in your main manuscript text for the figure legends--both for the main figures and supplementary figures
- please provide your tables in editable doc or excel format
- please use the [10 author names, et al.] format in your references (i.e. limit the author names to the first 10)

A. FINAL FILES:

B. MANUSCRIPT ORGANIZATION AND FORMATTING:

Sincerely,
Shachi

Shachi Bhatt
Executive Editor
Life Science Alliance
www.life-science-alliance.org

September 1, 2020

RE: Life Science Alliance Manuscript #LSA-2020-00771RR

Dr. Sheena D'Arcy
The University of Texas at Dallas
Chemistry and Biochemistry
800 West Campbell Rd
Richardson, TX 75080

Dear Dr. D'Arcy,

Thank you for submitting your Research Article entitled "Stoichiometry of Rtt109 complexes with Vps75 and histones H3-H4". It is a pleasure to let you know that your manuscript is now accepted for publication in Life Science Alliance. Congratulations on this interesting work.

*****IMPORTANT:** If you will be unreachable at any time, please provide us with the email address of an alternate author. Failure to respond to routine queries may lead to unavoidable delays in publication.*******

DISTRIBUTION OF MATERIALS:

Again, congratulations on a very nice paper. I hope you found the review process to be constructive and are pleased with how the manuscript was handled editorially. We look forward to future exciting submissions from your lab.

Sincerely,
Shachi

Shachi Bhatt
Executive Editor
Life Science Alliance
www.life-science-alliance.org